# Increasing Resilience of Production Systems by Integrated Design

**Steffen Ihlenfeldt [1,2], Tim Wunderlich [1], Marian Süße [1], Arvid Hellmich [1], Christer-Clifford Schenke [1], Ken Wenzel [1] and Sarah Mater [1,\*]**

[1] Fraunhofer Institute for Machine Tools and Forming Technology IWU, 09126 Chemnitz, Germany; steffen.ihlenfeldt@tu-dresden.de (S.I.); tim.wunderlich@iwu.fraunhofer.de (T.W.); marian.suesse@iwu.fraunhofer.de (M.S.); arvid.hellmich@iwu.fraunhofer.de (A.H.); christer-clifford.schenke@iwu.fraunhofer.de (C.-C.S.); ken.wenzel@iwu.fraunhofer.de (K.W.)

[2] Institute of Mechatronic Engineering, Technische Universität Dresden, 01069 Dresden, Germany

\* Correspondence: sarah.mater@iwu.fraunhofer.de; Tel.: +49-351-4772-2634

**Abstract:** The paper presents a framework for considering resilience as an integrated aspect in the design of manufacturing systems. The framework comprises methods for the assessment of resilience, supply chain and production planning, flexible execution and control as well as modular and skill-based methods for automation systems. A basic classification of risk categories and their impacts on manufacturing environments is given so that a concept of reconfigurable and robust production systems can be derived. Based on this, main characteristics and concepts of resilience are applied to manufacturing systems. As a lever of increased resilience on business and supply chain level, options for synchronized production planning are presented in a discrete event simulation. Furthermore, a concept to increase resilience on the level of business process execution is investigated, allowing manufacturing tasks to be rescheduled during runtime using a declarative approach to amend conventional business process models.

**Keywords:** resilience; production systems; matrix production; skill-based control systems; virtual commissioning

## 1. Introduction

### 1.1. Increasing Necessity of Production System Resilience

The anticipation, prevention, and mitigation of risks and their corresponding threats are an inherent part of decision making in economy and industry. It is not only the current pandemic situation that demonstrates how vulnerable supply chain networks are. International supply chain networks and customer markets require fast product transfer with minimized delays. The manifold system elements and aspects inside of manufacturing systems that are threatened by expected or unexpected events lead to chains of effects on factories and the incorporated manufacturing systems.

The following chapter provides an overview on the technological and IT-related counterparts for risk mitigation to increase resilience. Section 2 starts with the conceptual description of risk management as well as resilience evaluation with its specific application in the manufacturing environment. It also suggests the standardized automation pyramid and its corresponding reference architecture model 4.0 as a guideline that covers the relevant levels for production system description and the corresponding solutions that are presented afterwards. It is completed with specific sections of measures for resilience improvements, starting from the supply chain and enterprise down to field level.

Section 3 follows the same level-oriented structure and presents dedicated results that were generated by the methods and technologies described before. The paper closes with a discussion on the most critical aspects from the presented works, which are related to the

representation of complex and interrelated information as well as the derivation of matrix production as a future-oriented and resilient manufacturing system concept.

### 1.2. Technological Aspects and Requirements for Risk Mitigation

The established way of developing a production system was to design it according to a specified task and to increase productivity to a certain or required level. This led and still leads to static solutions in relevant domains such as process, structure, drive systems, and controller engineering [1]. Furthermore, data flows and energy supply were designed once and have remained more or less the same over the entire production system life cycle. A secondary interest lies in energy and resource efficiency, but this also leads to monolithic rather than modular solutions.

To effectively integrate resilience into the design of modern production systems, various changes must be considered, especially in the mentioned domains. As the most important aspect, the production system shall react to procurement, production, and personnel risks and therefore adapt to dynamic changes in order sequence and processing. Typical reasons are priority orders, changes in demands for products, shortages of raw materials, or shortages of personnel resources. Hence, the new generation of production systems must quickly adapt to changing processes and variations in throughput by using techniques for fast reconfiguration, commissioning, and fault tolerance. For example, [2] identified product and process modularity as key aspects to increase flexibility and hence improve the tolerance towards supply disruption. In order to realize such adaptivity for the whole production system, we propose leveraging modularity on all layers of hardware design, automation, and shop-floor software.

Furthermore, hardware and software design need to be intertwined by applying concepts such as decentralized controllers, self-description of modules, and the encapsulation of controller code into the modules [3].

Regarding connectivity and data exchange, standardized interfaces and data models should be established to support transparency and to complete traceability within a single production system as well as along the supply network. To accomplish this, the standard "Open Platform Communications Unified Architecture" (OPC UA) can be combined with linked data approaches [4] for modeling system structures, components, processes, and products as well as for linking them to related operating data.

Furthermore, manufacturing systems require a digital counterpart to enable fast commissioning, optimization, and controller code review in parallel to the production process. All of these aspects diffuse into a "framework for resilient manufacturing systems", which will be explained in detail in the following sections.

## 2. Materials and Methods

### 2.1. Risk Management and Resilience Concepts Transferred to Manufaturing Systems

Risk management as a measure and business discipline is typically known as corporate risk management, which focuses on accounting and financial reporting (see, e.g., [5]) or as a relevant step in project management processes (see, e.g., [6]). In order to evaluate risk probabilities and their impacts, the identification and classification of risks are inevitable. In a survey by Fries et al. [7], ten major challenges on future production systems were identified that can be subsumed into complexity (of supply networks and products as well as processes), changing customer behavior and expectations, market changes (globalization, volatility and increased competition) as well as impacts caused by politics, natural disasters, and unstable economics.

With reference to the aforementioned risk management, a structuring of risk categories is required. Various descriptions of system structure and meta-level representations have been developed in the context of factories (see, e.g., [8,9]).

A possible corresponding classification of risks refers to the depicted internal and external flows of production systems. Thus, we propose the separation of risks into the categories described in Table 1.

**Table 1.** Risk categories and corresponding cause–effect relationships.

| Category | Cause | Effect | Expression |
|---|---|---|---|
| Procurement risk | Political regulation | Raw material price increase | discrete |
| | Market shortage | Raw material price increase | continuous |
| | Complex supply networks | Lack of transparency | continuous |
| Production risk | Machine failure | Production stop | discrete |
| | Complexity of material flow | Increasing throughput times | continuous |
| Personnel risk | Increased competition | Fluctuation | continuous |
| | Changing expectations | Fluctuation | continuous |
| Sales risk | Increased competition | Price reductions | continuous |
| | Volatility of markets | Reduced forecast precision | continuous |
| Information (Technology)-related risk | Data inconsistency | Reduced forecast capabilities | discrete |
| | IT-security hazards | Data losses | discrete |
| Ecological risk | Climate Change | Temperature increase in production facilities | continuous |
| | | Increased probability of extreme weather events | continuous |

In industrial practice, enterprises are possibly confronted with combined threats of several risk categories. In addition to that, it should be highlighted that each of these risk categories is translated into an economic evaluation, as corporate decision-making ultimately depends on monetary performance metrics. Thus, each risk category requires translation into specific cause–effect relationships in order to operationalize risk assessment and to elaborate measures. Moreover, the effect of one risk may even express itself in further categories. Conversely, causes may be relevant in several risk categories. Furthermore, the risk effects can be separated according to their time-related behavior. According to this, Table 1 illustrates a typical specification of risks.

With regard to production systems, these risks may cause external and internal turbulence, increasing the necessity of short adaption and transformation times. This leads to the necessity of flexibility and the quick configurability of elements and processes in manufacturing facilities. From a system-based point of view, an overall production system may be divided into several subsystems of interchangeable and reconfigurable components based on standardized interfaces and a modular structure. Figure 1 depicts this system concept as a framework to anticipate as well as to mitigate risk impacts and to generate a common understanding of subsequent sections.

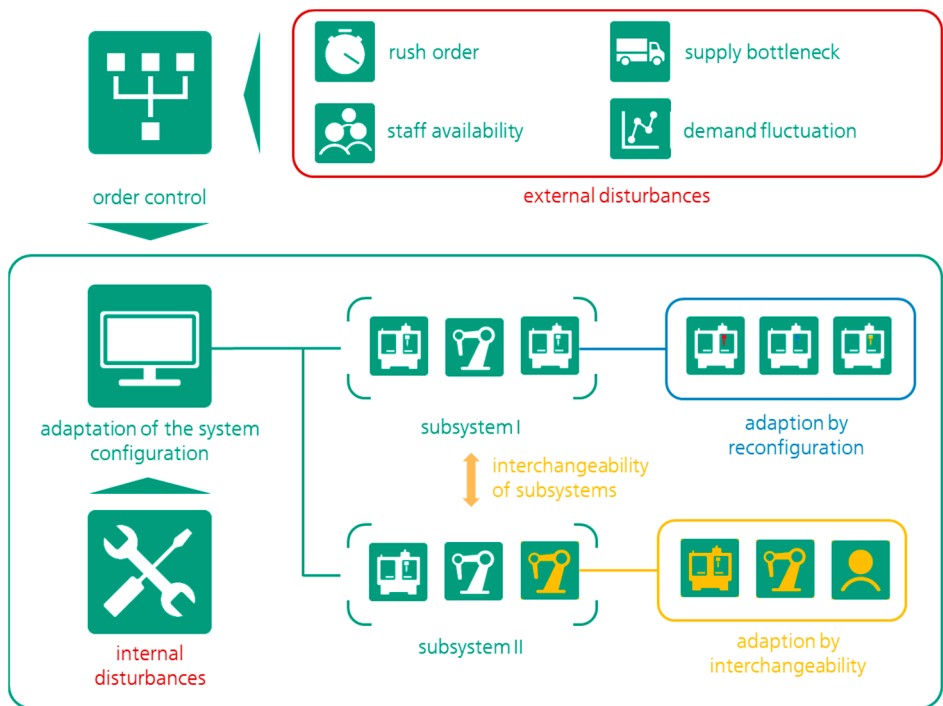

**Figure 1.** Concept of a situationally optimized production system for increased resilience.

Resilience as a feature of systems covers multiple aspects and dimensions related to the prevention and mitigation of risks. A conceptual distinction between "engineering resilience" ("efficiency of function") and "ecosystem resilience" ("existence of function") is traced back to Holling (1996) [10]. The first represents stability in the sense of efficiency, continuity, and predictability in order to generate fail-safe (technical) designs and to always remain close to a targeted state of equilibrium as a given measured variable. In contrast to this, the second form describes resilience as persistence, change, and unpredictability, in which a system can flexibly adapt to new conditions. In this case, resilience is measured by the amount/extent of disturbance that can be absorbed prior to system change. Hence, ecosystem resilience often refers to the assessment of complex social and ecologic systems.

Fischer et al. (2018) [11] mention that resilience as a term is used in many variations, depending on the scientific area. With a focus on urban environments and systems they developed a mathematical resilience framework for evaluating and comparing different prevention and reaction strategies for hazardous situations on a quantitative basis. The derived resilience cycle comprises the phases prepare, prevent, protect, respond, and recover and is based on a study published by the German Academy of Technological Sciences (acatech) [12]. Thus, this cycle expands the formerly established social resilience cycle, which comprised the four phases of preparedness, response, recovery, and mitigation, that was described by Edwards (2009) [13]. Figure 2 illustrates the described resilience types and cycle phases in a schematic manner, applied to the impact of hazards on production system performance.

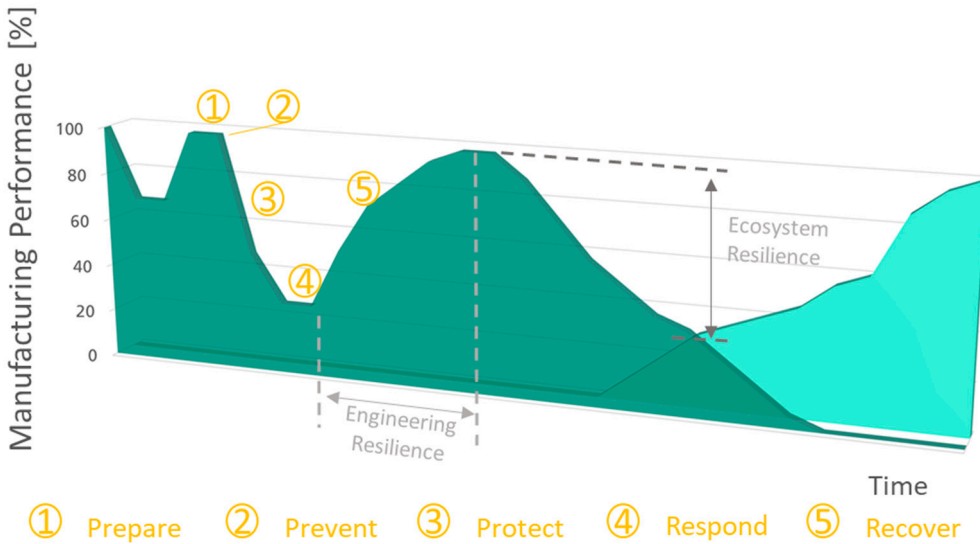

**Figure 2.** Differentiation between resilience types and resilience cycle phases.

The illustrated time series helps to allocate terms and methods that aim to reduce the effect of internal and external disturbances on manufacturing system performance. In an exemplary scenario, the initial performance reduction—caused, e.g., by supplier breakdown—may lead to the decision of preparation strategies such as (1) the improved synchronization among supply chain partners. In case of assumed bottlenecks, prevention methods such as (2) the increase of incoming stocks may be performed to reduce the impact of interruptions in material supply. Protection methods (3) may comprise strategies of supply chain management and production scheduling in order to mitigate performance losses due to supply failures. As a response (4), product components are substituted and lead to the necessity of system reconfiguration due to adapted production processes. The corresponding recovery phase (5) essentially affects profitability and competitiveness. However, market demand for the fictitious product may steadily decrease, caused by changing customer demands and may finally lead to the transformation of an overall production system, which is illustrated by shifting from dark to lighter green. Based on this classification, the actions and reactions during dangerous impacts can be sorted along their time-related occurrence. However, in order to develop a structured portfolio of measures, the appropriate model as a guideline is required.

The automation pyramid is a well-established model for factory automation [14]. In accordance with the recent trend of Industry 4.0, this model was extended to a three-dimensional expression, including more aspects than the level of automation. The resulting RAMI 4.0 (Figure 3) includes elements and IT components in a layer and life cycle model. It tries to decompose complex processes into modules and adds aspects of data privacy and IT security. RAMI 4.0 is intended to elaborate the understanding and discussions of all participants involved in Industry 4.0 [15]. Thus, it may also serve as a base model for the definition and systematization of resilience-increasing measures.

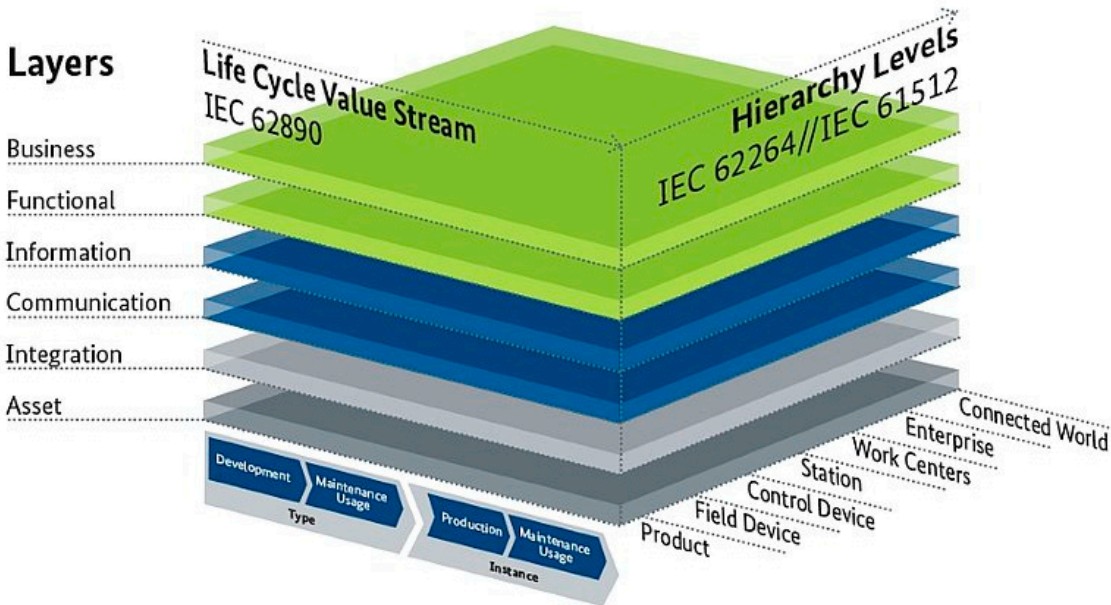

**Figure 3.** Reference Architectural Model Industry 4.0 [15].

Related to the described resilience cycle phases, the subsequent section contains simulation-based options that mainly focus on phases 1 to 3 on an enterprise- and supply chain level. Afterwards, the necessary concepts in information modeling and control for fast reconfiguration are described.

*2.2. Concurrent Supply Chain and Production Planning*

In order to investigate disturbances on production systems and the accompanying supply chain networks, material flow simulation provides appropriate options to study system behavior. According to Guideline 3633, Sheet 1 of the Society of German Engineers (VDI) [16], simulation involves the recreation of a system with dynamic processes allowing the investigation with experiments to generate findings that can, in turn, be transferred to the real system. Thus, it becomes possible to investigate the temporal behavior of the respective modeled system and to derive corresponding statements. For the investigation of processes, discrete-event flow simulations are often used, in which the time-related behavior of the system is represented by successive events of a processing list and the associated state changes.

In an exemplary use case, a model was implemented in a Tecnomatix® Plant Simulation, which contains its own user interface to select production scenarios depending on the processing status of the implemented orders. A truck picks up the finished orders from the last day based on the first-in-first-out (FIFO) principle. Depending on the destination of the products, there is a separation between two different retrieval strategies. Inland locations allow partial deliveries, whereas abroad destinations require complete order fulfilment in one iteration. Disturbances can be turned on and off for the study of different scenarios with a simple button. This means that once the orders run through as planned (deterministically), there will be no delays in production. If the button is activated, disturbances with a defined mean time to repair (MTTR) are switched on, stochastic effects generate impacts on processes, and therefore, delays and unexpected events can occur.

In its standard version, the modelled factory works in a 2-shift system. A total of two hours before end of production, a message is sent to the user to make a decision on how to proceed with further production. The user can base this decision on the status of the planned orders and the orders being processed at that time.

In doing so, the user can select a scenario (e.g., start next shift) in the interface. This means that a third shift is run because the user assumes that the orders cannot be completely

processed until the end of the second shift. In general, the user has three options as coping strategies for disturbance-related delays:

- Option 1: Start third shift (as described above);
- Option 2: Increase production speed by a certain percentage. The default value in this case is 15%. Hence, orders may be completed by 10pm after all or will continue to be processed the next day;
- Option 3: Continue as planned; in this case, it could be the case that the orders become behind schedule and have to be processed within the next day.

The implementation and operationalization of the described options in a short-cycled and complex production environment necessitates the sophisticated description and modeling of system elements and their corresponding production processes. Therefore, an appropriate modeling approach and the derived process generation functionalities are described in the subsequent section.

### 2.3. Modelling and Execution of Resilient Production Processes

Mitigating risks and increasing resilience on the level of workflow management and process execution requires that an erroneous process can be returned to a defined state and—if possible—the defined goal of the workflow can still be reached. For example, if a single machine fails within a production process, it is in the interest of the plant operator that the error is remedied as soon as possible, reducing the loss of production and therefore reducing costs. Additionally, it might be feasible to reschedule single tasks within the production process in order to reduce downtime further. With the example of a model machine that allows for rescheduling the tasks performed by its modules, we examine how resilience can be provided on the level of process execution. On the one hand, this approach requires an amendment to the models used to describe production processes, and on the other hand, the information flow between the involved components of the Manufacturing Execution System (MES) needs to be taken into consideration.

Classic business process modelling consists of a sequence of activity nodes, which are intertwined with decision nodes, allowing for as much as flexibility as possible to be considered at compile time. In case erroneous events take place during the process execution, those events must be an explicit part of the process model. Consequently, an equally explicit mitigation strategy is required to be modelled at compile time as well, meaning that even "the unforeseen" must be foreseen. However, several approaches have been explored to allow for more flexible process execution and to provide resilience on the level of process execution in the domains of software engineering [17], emergency management [18], logistics [19], and cyber-physical systems [20]. Notably, [18] advocate a declarative approach, annotating single activities with "preconditions and effects", allowing for a dynamic workflow generation by utilizing these annotations as constraints between those activities.

However, the domain of cyber-physical production systems (CPPS) imposes further conditions since tasks executed by production machines or human operators usually affect artifacts existing in the real world—such as work pieces—and thus manipulate their state. Furthermore, these artifacts may be impacted by influences that are outside of the scope of the defined workflow, leading to cyber-physical deviations that need to be addressed during process execution [21]. Therefore, we propose the generalization of the declarative approach used in [18] by considering the condition of the artifacts and by utilizing their context for applying constraints on the tasks within the production workflow. For this matter, we use the definition of "context" provided by Dey and Abowd (2000), who define context as "any information that can be used to characterize the situation of an entity" [22]. They further define an entity as "a person, place, or object that is considered relevant to the interaction between a user and an application, including the user and applications themselves", but given the nature of CPPS, interactions do not only take place between users and applications, but also in-between applications. For this reason, we suggest not limiting this definition to interactions in which a user participates.

Within the research project "RESPOND", we evaluate the declarative approach outlined above with the aid of a model machine consisting of four modules. These modules are traversed by a work piece made of aluminum (approx. 65 mm $\times$ 20 mm $\times$ 3 mm) in a skid on a conveyor belt. The four modules and their respective tasks are as follows:

1. A drill that drills a hole into the work piece;
2. A milling machine that mills an engraving into the surface of the work piece;
3. A camera that measures the diameter of the drilling;
4. A knuckle joint press to manually press a steel ball into the drilling.

Furthermore, a human operator is responsible for loading and unloading the machine. In order to execute task 4, tasks 1 and 3 must have been executed successfully before. However, task 2 is independent of the other tasks and is only crucial for the success of the whole workflow. This means that if task 2 fails (e.g., due to a malfunction of the milling machine), the other tasks can still be executed and task 2 can be performed later, provided that the defect is remedied. Furthermore, executing the automated tasks (drilling, milling, measuring) requires that the work piece is in the skid of the conveyor belt, while the manual task requires that the work piece has been taken from the skid into the knuckle joint press.

By annotating the activities of the process with the required and returned context data, it is possible to verify whether the process is valid. For example, if an activity requires that the work piece be located inside the skid, a preceding activity must have returned "location = skid". This standard process annotated with the required and returned context data is depicted in Figure 4.

Furthermore, these annotations can be utilized to dynamically provide resilience on the level of process execution in case one of the modules fails to perform its designated task. The use case that we want to examine represents a temporary defect of the milling machine, which is supposed to be remedied by a human operator. This resilience can be provided in the form of a modified workflow, which is generated ad hoc upon process errors. For this matter, we propose a Process-Planning Engine (PPE), which can be accessed by the Manufacturing Execution System (MES) and that is able to obtain information from the process itself as well as a context model of the plant.

The information flow outlined above requires a suitable architecture, in which contextual information is available in real-time to the PPE. This use case is explored in [21], where an architecture consisting of four layers allowing for real-time communication between the involved components is proposed. One of these layers (the so called "RESPOND infrastructure") is responsible for communication via an event bus. Payloads delivered via this event bus include, e.g., sensor data, commands for actuators, or messages sent by components upon their registration. Besides the event bus, components communicate via a peer-to-peer connection. Within this architecture, the proposed PPE can be represented by the "Process Healing" node.

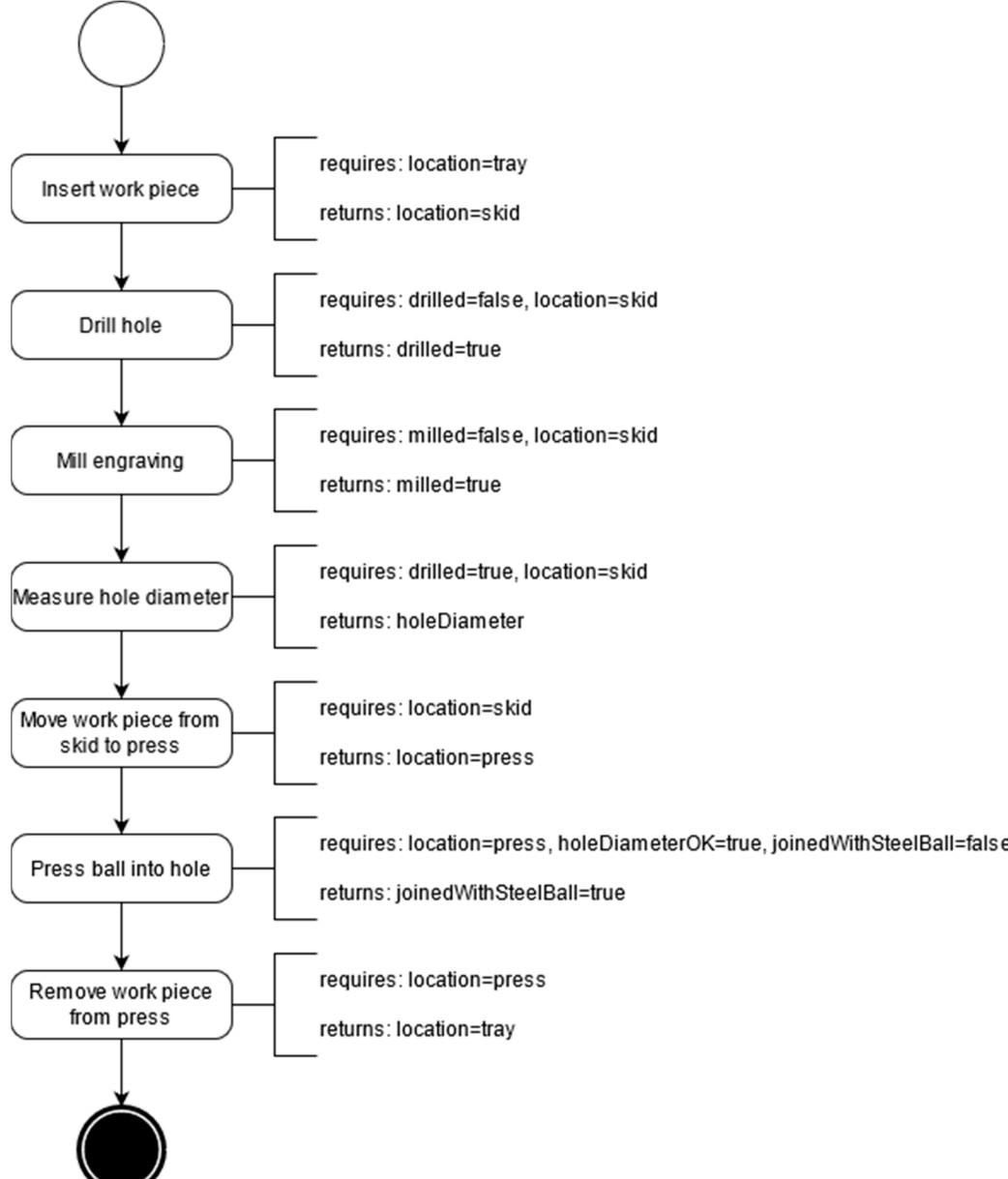

**Figure 4.** Default process of the model machine described in Section 2.4. The activities are annotated with the requirements that the work piece must meet in order to allow for an execution of the activity ("requires") as well as with the context changes that are applied to the work piece after the execution of the activity ("returns").

Based on this architecture, a sequence diagram displaying the information exchange relevant for resilient process planning is shown in Figure 5. As soon as a fault in the current process is detected, a message containing the IDs of the faulty process, the blamable activity, and the involved agents is sent to the PPE. For this matter, the origin of this error message is considered a black box; however, it is conceivable to provide these error messages by means of a Complex Event Processing (CEP) engine. In the architecture proposed by [21], the CEP engine is represented by the "Process Analysis" node.

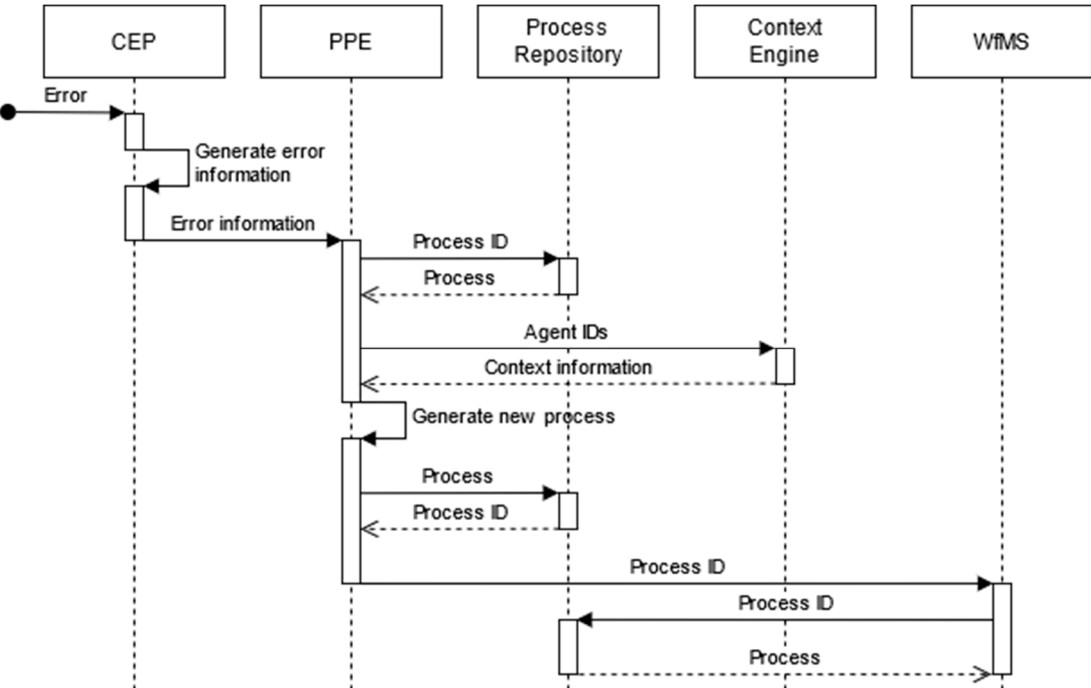

**Figure 5.** Sequence diagram showing the information flow in order to generate a so-called "healing process" after an error has been detected.

Based on these IDs (process, activity, agents, artifacts), the PPE requests the process repository for the process model and the context engine for context information about the involved agents and artifacts. Assuming that the process model is amended by semantic information about the artifact states as shown in Figure 4, the "required" and "returned" states can be summarized as "modifications" and can be mapped to the necessary skill profile of the involved agents. Based on these modifications and skill profiles, two general approaches can be conducted (Figure 6):

1.　Adjust the faulty process by replacing the faulty activity with another activity (or sub-process) that yields the same outcome;
2.　Transfer the faulty activity to another agent that corresponds to the required skill profile.

As soon as the PPE finds a solution for the faulty process, the newly generated process is pushed to the process repository and its ID is published via the event bus to be read by the workflow management system (WfMS), which, in turn, retrieves this process from the repository and then starts it.

In the example of the workflow depicted in Figure 4, the task that should be performed by the milling machine would be rescheduled to the end of the workflow, and a human operator would be instructed to repair the machine. However, the milling machine requires the work piece to be in the skid on the conveyor belt, and at the end of the default workflow, the work piece is unloaded from the machine and put into the tray. Therefore, it is necessary that the rescheduled milling process is padded with further activities that guarantee the required artifact location, i.e., putting it into the skid before the milling and taking it from the skid to the tray afterwards. The final activity was not part of the original process model, so in order to add this activity to the rescheduled workflow, the PPE needs to query the process repository for suitable activities or sub-processes based on the required context modification. This modified workflow is depicted in Figure 7.

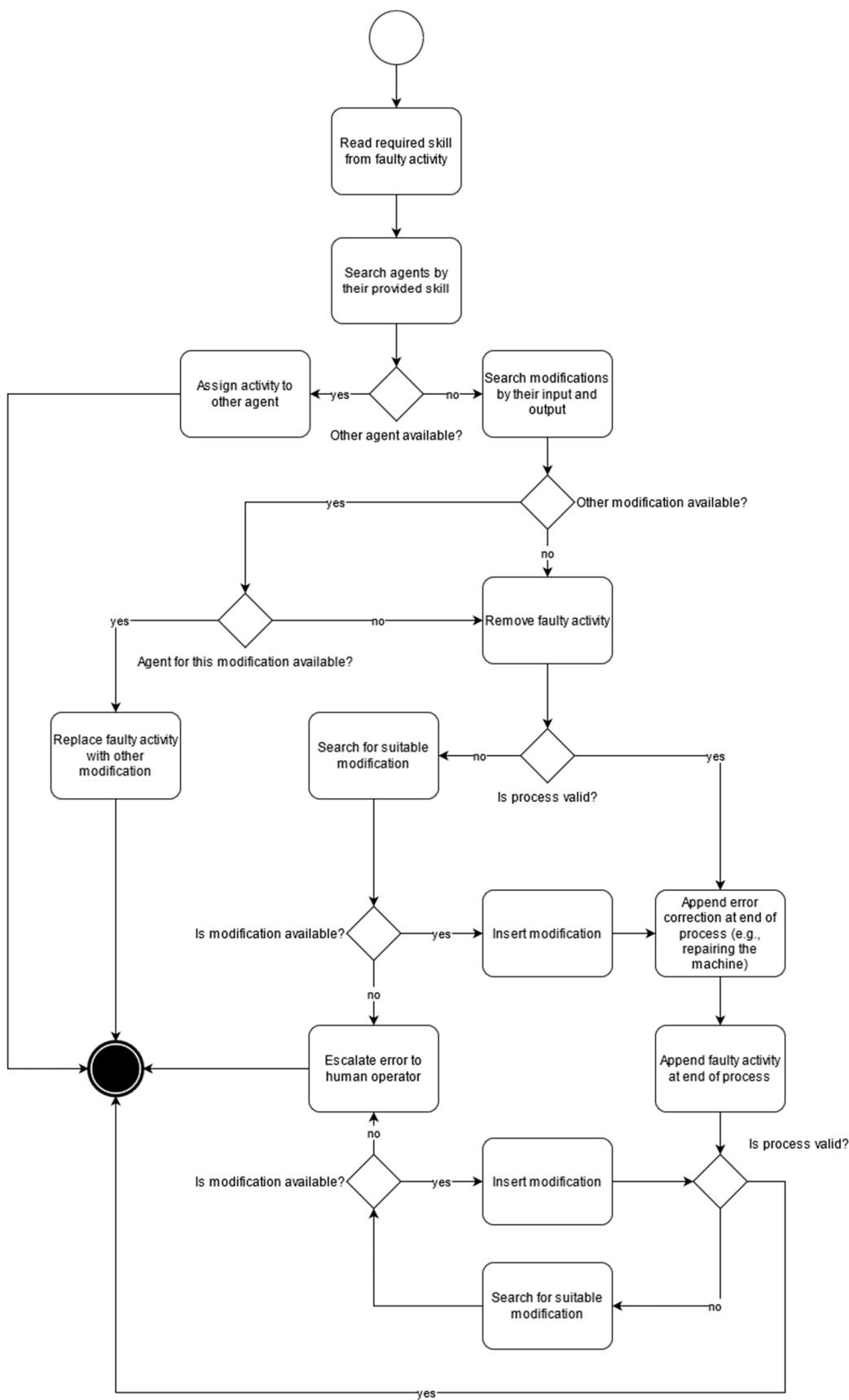

**Figure 6.** Basic workflow inside the Process-Planning Engine to generate a new process upon an error due to a defective machine. A set of "required" and "returned" conditions is summarized as a "modification".

*2.4. Rapid Response to Changing Requirements Using Skill-Based Control Systems and Virtual Commissioning*

To implement the methods of workflow management and process execution to increase the resilience of production systems, it is necessary to increase flexibility and modularity on the station level and below. This can be achieved by designing the control device, field device, and product accordingly. The modular design of the control device reduces the development time and expenses even for smaller batch sizes. In addition, this modularity enables flexible and individual adaptation of the control device to the application. The focus of this section lies on the control device and its interaction with a digital representation of the production station, represented as digital twin. This combination is a key enabler in bringing production systems towards more resilience in production and is also beneficial in the development phase. Details will be given in two subsections for the controller and digital twin.

Modular and flexible production stations require new design [23] and new ideas of power and information supply [24] as well as a new paradigm of PLC programming. Only if the flexibility and adaptability in hardware is transferred into flexibility in the design of automation solutions can a true and holistic flexibility in manufacturing be reached. However, state-of-the-art programmable logic controllers (PLC), which are the common solution, e.g., for robot cells, are a cyclic processing comprising the steps:

- Input scan (reading all inputs and memory states);
- Execution of a problem-oriented automation program (PLC-program) to generate output and memory values;
- Output update (writing values to outputs and memory).

Up until today, the core of the PLC program has been defined by the automation task. Until now, this has usually been developed, implemented, tested, and maintained on a task-specific basis by an automation technician of the machine/unit manufacturer [25]. Due to the fact that the task defines the code, necessary modification, adaptation or addition of command sequences, and positions, process sequences are usually not easily possible, even for minimal changes to the automation task. The effort required for programming, testing, and commissioning control software is growing disproportionately with the increase in the scope and complexity of control functionality [26]. In addition, monolithic, task driven programs are mostly only changeable by specifically educated PLC programmers, which often result in long maintenance breaks. Hence, this paradigm must be changed due to the stated request for increased flexibility and modularity of production to realize an extended resilience.

An alternative is the plug-and-produce concept. Although this was originally developed for hardware and connectivity in automation systems, it enables the flexible configuration and partial self-organization of production processes at the runtime of the system. Individual functional components can be combined and/or exchanged in a flexible manner in order to adapt the production system to changing products or boundary conditions. The basic idea of plug-and-produce is that hardware components make their functions available based on a self-description, including all of the necessary information for the higher-level automation system. Thanks to a uniform interface, new components can be easily connected and used by the control device [14]. Several solutions are already available, mainly using the OPC Unified Architecture (OPC UA) standard for data exchange [27]. This was developed as a platform-independent, service-oriented architecture. However, most of the existing solutions are still manufacturer dependent or limited to several, specified use cases [28].

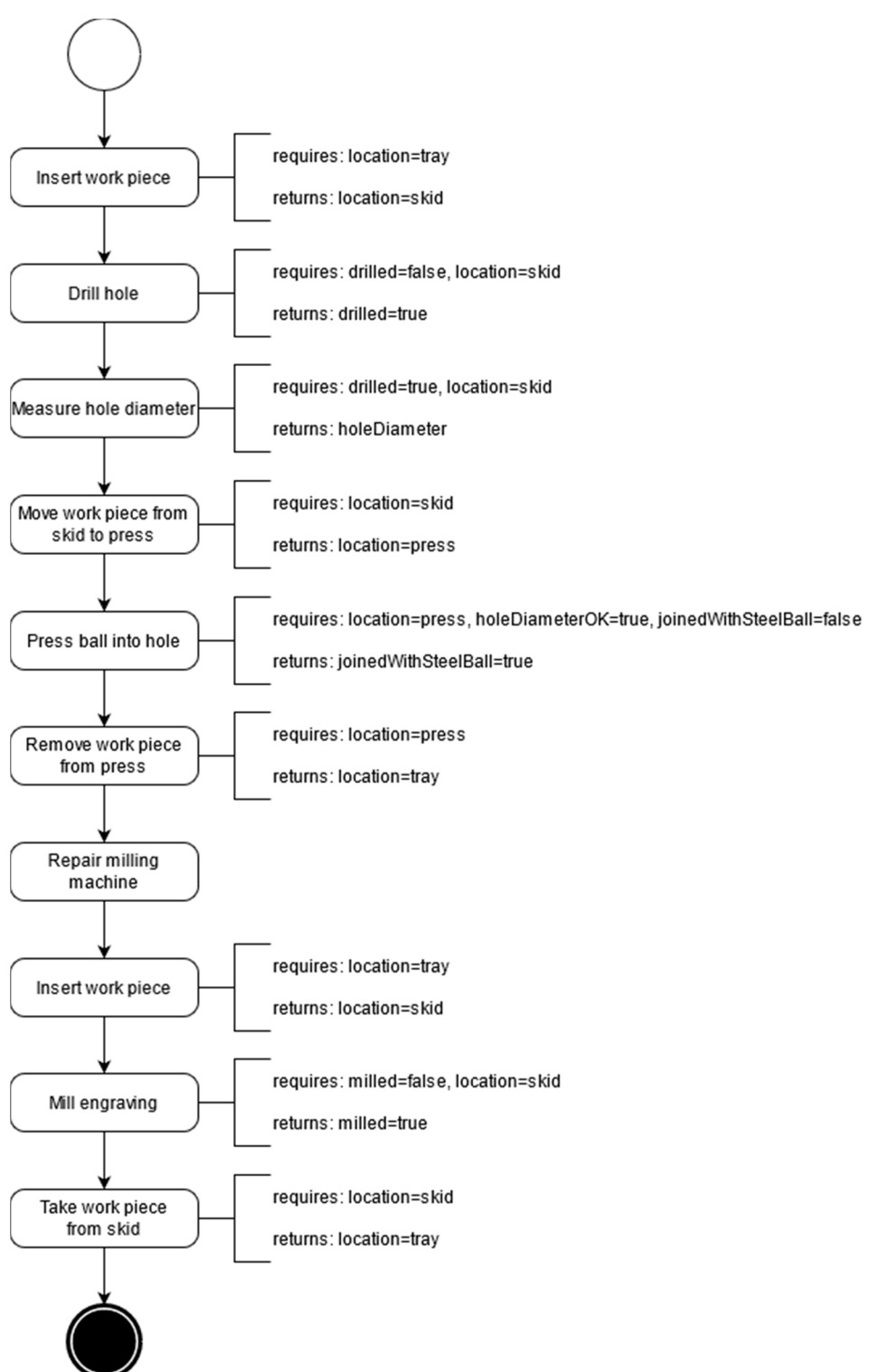

**Figure 7.** The original workflow has been rescheduled to account for a defect of the milling machine. The task "Mill engraving" has been relocated to the end of the workflow and has been padded with "Repair milling machine", "Insert work piece", and "Take work piece from skid", so the constraints imposed by the required and returned context data are still valid.

One basis for implementing the plug-and-produce concept at the level of control devices is to structure the functional components of an automation solution into modules. This ensures that modules can be easily exchanged and arranged according to the automation task as well as to the hardware representation.

This also triggers the main disruption in the programming of control devices: the developed code is no longer task-driven, but skill-driven and modular.

The most important features of this controller programming approach are uniform interfaces and the provision of parameterizable module functions in specific skills. Afterwards, the functions can be orchestrated freely for the automation task by a higher-level system, whereby each module provides the necessary function descriptions, requirements, and boundary conditions [13].

The strengths compared to the established architectures lie in the free reconfigurability, interchangeability, and reusability of functional components at the control level. The concept of skill-based control can be derived from a product-process-resource model (see Figure 8).

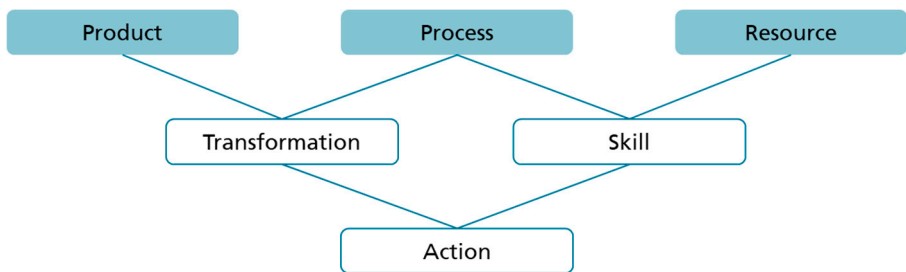

**Figure 8.** Capability concept based on product, process and resource inspired by [29].

The benefit to the user is that the PLC programming is transformed from a programming effort to a combination of predefined skills and their parametrization. Hence, the controller is not only equipped with a specific program to solve one specific complex automation task (Figure 9, left), but bases it on a large set of modular basic skills, covering all of the possible or senseful abilities of the manufacturing unit. Here, skills can be parametrized and combined to jobs, leading to the complete automation task (Figure 9, right).

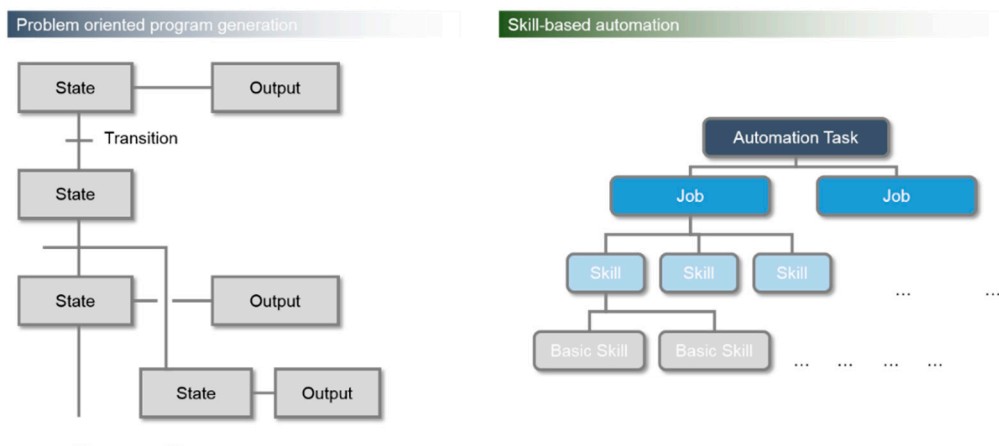

**Figure 9.** Transformation of problem-based to skill-based automation.

To give an example, handling, measuring, orienting, and loading/unloading are regarded as jobs. All of the jobs consist of a sequence of skills, such as movement, opening/closing a gripper, call to a camera. The skill "movement" is furthermore a combination of basic skills such a move linear or move circular (Figure 9, right). The manufacturing cell is equipped with all hardware modules and a complete skillset. This skillset is taught and programmed by the automation technician but remains open for a variety of applications. The solution is extremely flexible in further applications and in individual utilizations. Further steps such as combining skills to jobs and programs can be conducted by skilled workers and do not require automation technicians.

After a basic commissioning and software test, the machine operator can combine the skills and jobs to an automated process sequence or adapt the given sequence to a new setting with support of a graphical user interface (GUI).

The fast commissioning as well as the software tests for the skill-based functions can be realized by virtual commissioning using a digital twin of the production system. Digital twins are already widely used by machine and plant manufacturers to virtually plan and optimize the most complex production machines and plants during their development [30]. Along the development process, the digital twin can be used for the design of the different domain-specific subsystems and to evaluate the dependencies between them. As such, at the end of the construction phase, there is a nearly exact digital twin of the machine that can be used for the development of the control functions while the hardware is in realization [31]. The virtual commissioning of control systems allows for a reduction of the commissioning time of the whole machine and results in a faster production start as well as reduced costs for the engineering process [32]. Figure 10 shows the timeline for the conventional procedure for the development of a system and in comparison, the timeline for the same procedures with virtual commissioning.

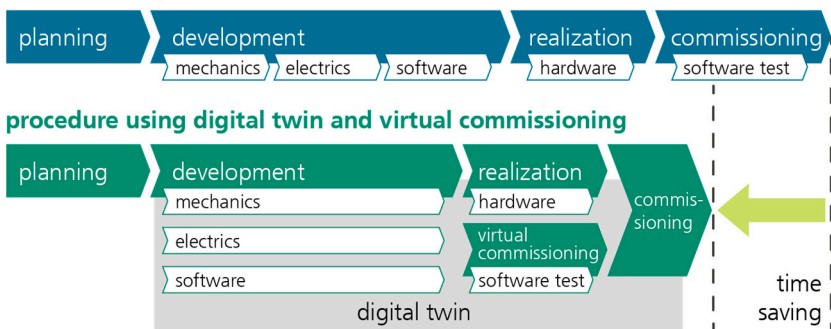

**Figure 10.** Reduction of commissioning time by virtual commissioning.

For virtual commissioning, machine and plant models are developed that correspond to their real counterparts in terms of their interfaces, parameters, and operating modes. With the help of these virtual systems, realistic test and commissioning situations can be simulated, including all control functions, whereby the control system can be operated and examined on the digital twin in the same way as on the real machine. Although domain-specific behavioral models are occasionally used today in the production phase for control and regulation functions, the comprehensive system simulation models from engineering are usually not adopted in the real operating phase. In the context of the resilience of production systems, these models in combination with the possibilities of the virtual commissioning contributes to various aspects of resilience enhancement.

Increasing the resilience of production systems not only requires the early recognition of the internal and external events that lead to disruptions in production processes, but also strategies to respond to them flexibly. For example, it is necessary to evaluate a suitable alternative in the event of a production equipment failure. In addition to manufacturing products of the same quality, the objective may also involve meeting deadlines or changing input materials, which can lead to necessary changes in the production processes. The adaptation of the production system must be planned and examined for its feasibility. If the machinery of a component manufacturer, for example, is virtualized, a virtual commissioning system can be used to assemble machines and systems from a library [33]. With this modular system, it is possible to quickly assemble new configurations for manufacturing systems, plants, or even entire factory halls in the event of changing boundary conditions and to examine them regarding defined optimization criteria. In terms of anticipation, such a configurator can be used to simulate events such as the failure of production equipment and to plan the corresponding reactions and validate them on the virtual representation.

This way, it is possible to react much more quickly in the event of an occurrence and thus shorten the downtimes of production systems.

Today, up to two thirds of the control software for manufacturing systems is used for error detection and handling [34]. The testing of the methods used for this can only take place after the hardware has been realized. Only then can the corresponding errors be provoked, and the reaction of the control system can be validated. In addition, errors that lead to the destruction of the system, impairment of the machine environment, or even injury to the machine operator cannot be checked on the real system. With the digital twin, on the other hand, it is possible to examine these error patterns in an early phase of the system development, even before its realization, and to incorporate the gained knowledge into the development of the system components and control system. In this way, the robustness of systems in the face of faults can already be increased during their development.

## 3. Results

### 3.1. Increased Business Resilience by Means of Concurrent Supply Chain and Production Plannning

The planning strategies, which were already indicated in Section 2.3, provide exemplary levers for increased predictability among connected supply chain partners and on the business level that are specifically implemented and investigated with discrete event simulation tools. Therefore, a scenario-based analysis of production order fulfilment is possible in a quantitative manner. The exemplary production plan comprises the following orders:

- 10 orders on day 1;
- 7 orders on day 2;
- 8 orders on day 3;
- 9 orders on day 4;
- 5 orders on day 5.

Filling additional outgoing stock is not taken into account in the scenario. The differences between the deterministic and stochastic forecast of the production process can be determined, as depicted in Figure 11.

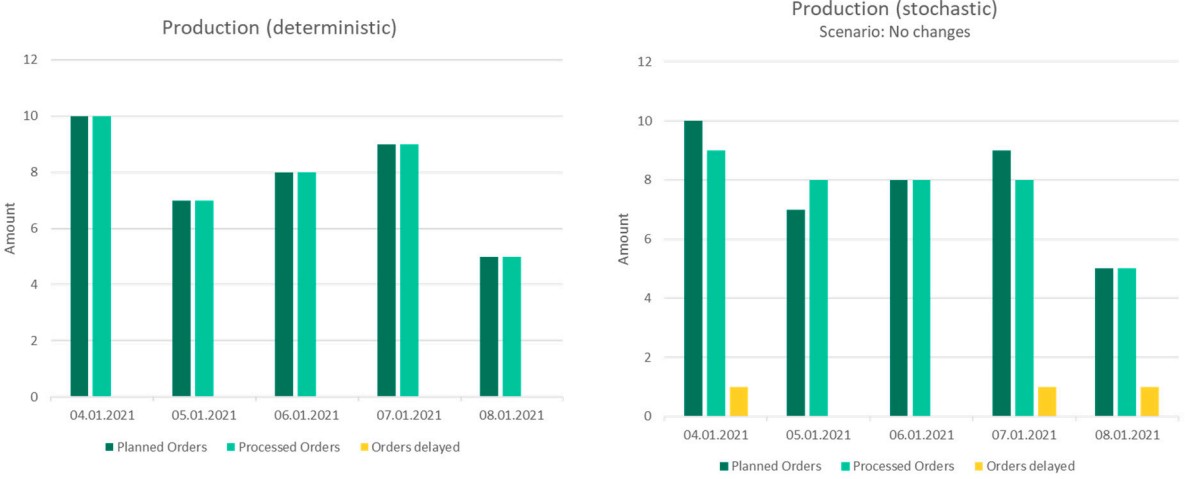

**Figure 11.** Comparison of influence stochasticity on order fulfilment.

In this case, the stochasticity refers to an additional consideration of MTTR, i.e., down times can occur unexpectedly, based on a predefined probability distribution. It becomes clearly visible that a reduction in processing performance at a distinct point in time can only be mitigated by increasing the production rate on a following day; otherwise, order delays are dragged along succeeding days. With the aim of synchronizing supply chain

control and internal production management, further scenarios (based on the stochastic start solution) are investigated. Hence, Figure 12 shows comparative results if production speed is increased or an additional shift is started.

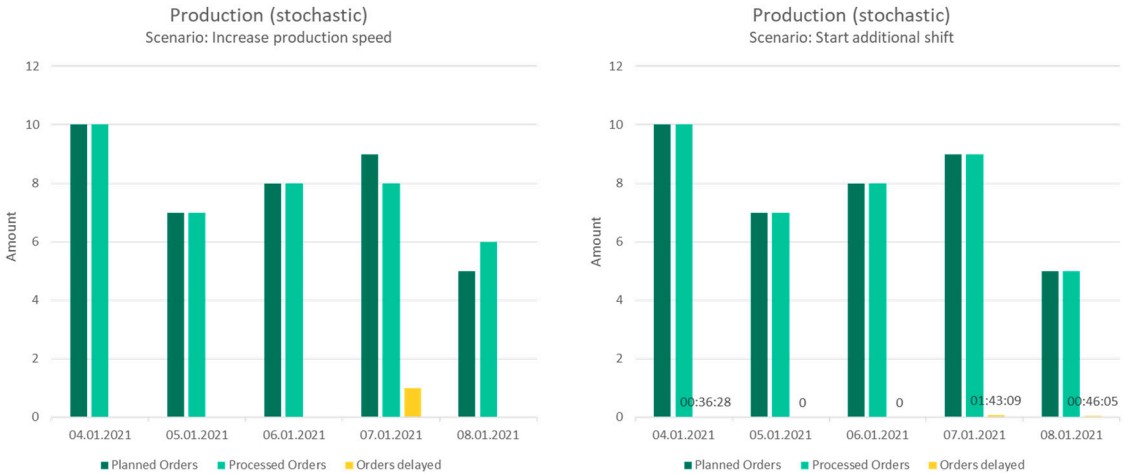

**Figure 12.** Comparison of derived coping strategies on order fulfilment.

It becomes visible that the increased production speed on the left-hand side helps to reduce delays during the planning cycle. However, order delays may still occur between single working days but can be recovered shortly afterwards. A further shift leads to an enormous increase of production capacity, and therefore, delays do not occur in the described scenario. The additional times on the right-hand side indicate how long it takes to fulfill the order during the third shift. It should be highlighted that the outcome in this case may not be universally transferred to every use case since the profitability and efficiency of each scenario heavily depends on the underlying parameters (e.g., production rate increase, personnel costs during night shifts).

However, the applied discrete-event simulation provides the base for improved synchronization between supply chain demand and internal operations. With reference to this, the corresponding implications on information architecture and manufacturing cell design are subsequently presented.

### 3.2. Architectural Implications Due to Process Resilience

With the declarative approach explained in Section 2.3, we examined how resilience can be provided on the level of process modelling as well as execution and explained how this approach fits into the architecture provided in [21]. However, adapting and generating processes at run-time does not only require the WfMS to be capable of reacting to these dynamic changes (i.e., terminating the current process and launching the "healing process"), but also includes assigning activities to specific agents in a flexible way. Therefore, these premises impose requirements on both the station level and the system architecture, i.e., the PLCs used to control machines, robot cells, etc., need to be addressable in a flexible way without causing an overhead in reconfiguration. A possible solution to this problem is explained in Section 2.4, with the approach of PLCs organized by skill profiles. On the one hand, this allows for a H data source that can be utilized during the process generation in order to find possible solutions to erroneous workflows, and on the other hand, once a possible solution is found, modular PLCs allow for a rapid reconfiguration on the station level.

Furthermore, some risk mitigation strategies involve human intervention: a defective machine or module might need to be repaired by a technician, a work piece needs to be loaded manually into the machine, or an error report needs to be acknowledged. Consequently, the system's architecture must support integrating humans into these processes as well, e.g., in terms of a role-based user management. When a human agent is notified

about a possible task ad hoc, further consideration is required regarding the presentation of this information. Besides the technical details of the implementation of such notifications, the necessary information to be presented to the human that is required to execute the assigned task needs to be determined. Furthermore, resilient processes that are (partially) executed by humans also raise issues in terms of cognitive ergonomics and human factors, especially if the skill-based approach used to manage production machines, robots, etc., will be extended to human operators.

To sum up, we highlight the following implications to be considered when integrating resilient production processes:

1. Modular management of workflow agents;
2. Adaptive and flexible workflow management;
3. Human-centric presentation of information.

### 3.3. Modular Manufacturing Cells for Problem-Independent Production

Figure 13 shows a GUI for a robot cell to manipulate, measure, and stack up automotive parts. The cell consists of a KUKA robot and four stations, each realizing different process steps. The left part shows the parametrized job list followed by a specification for each job. A visualization in the right part provides an interaction with the operator and visualizes the abstract robot actions associated with the manufacturing layout and task. Here, the programming paradigm interacts with factory facility planning when the state-actual model is incorporated in the GUI. In the middle part, control buttons allow for the PLC to be connected to, the download of the program to the PLC, and the running of specific jobs.

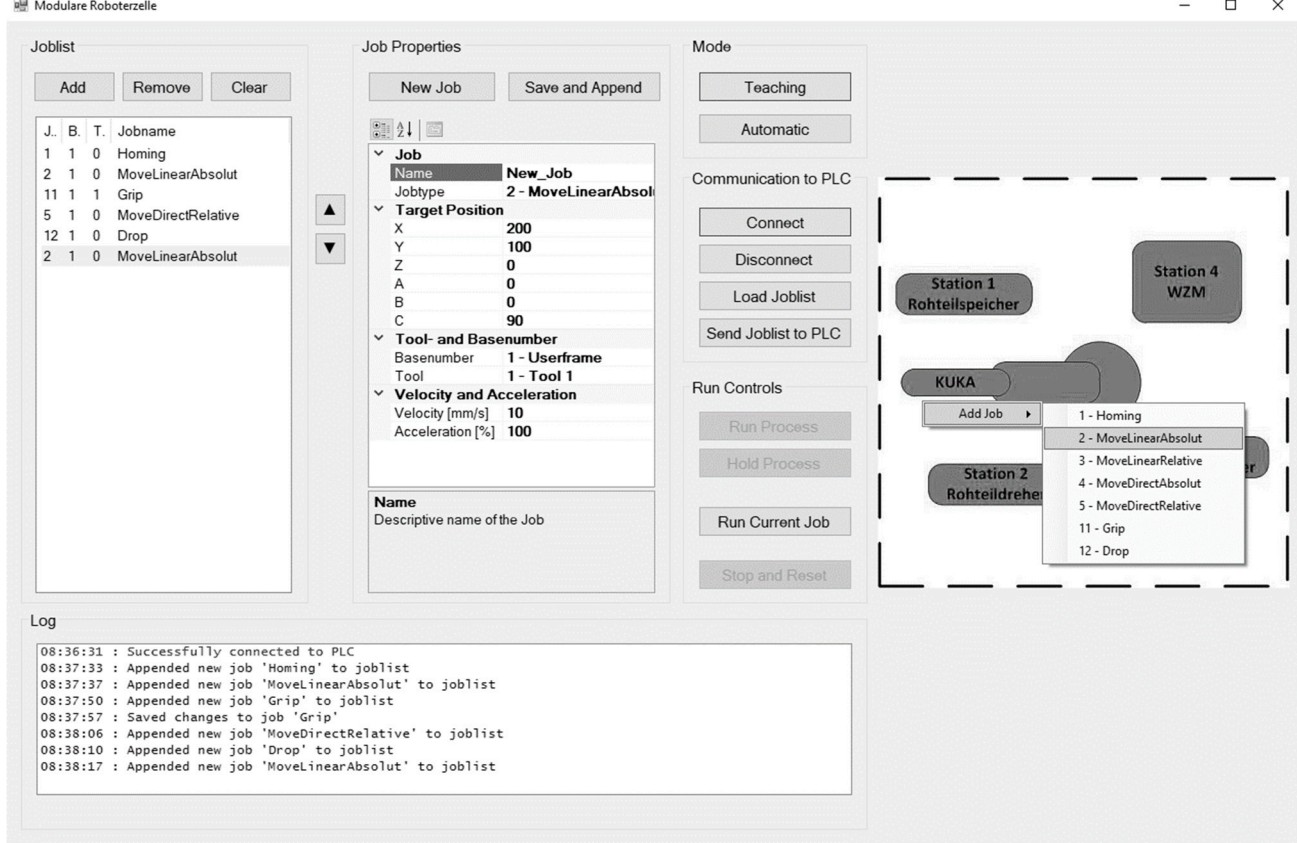

**Figure 13.** GUI of the skill-based automation for a flexible robot cell.

The new parametrization paradigm uses flexible robot jobs, parametrized by a GUI and subsequently downloaded into the PLC. The program, job/skill parameters, and the

chronology of skills can be controlled, adapted, and reorganized through a GUI. After basic commissioning of the complete skillset, no automation specialist is needed anymore. Job and process flow adaption and testing can be realized within a few minutes. Users without PLC or robot programming knowledge can implement changes, and the risk of errors in programming is reduced.

Besides the aforementioned robot cell for manipulating automotive parts, another robot cell for the highly flexible loading and unloading of machine tools was automated based on the new parametrization paradigm.

For the test of control functions and the evaluation of methods to increase the resilience of production systems, a model, shown in Figure 14, of a small production cell for additive manufacturing with machining finishing was created. The material flow within the cell is realized by different handling robots and a conveyor belt that transports the produced semi-finished products between the processing stations. For the finishing of the products, two machines for milling are modeled. One is a conventional cutting machine tool, and the other one is a robot with a milling spindle as tool. The coordinate transformation of the robot enables cartesian machining in three axes. Both machines are controlled by a NC controller so that the same NC code can be executed.

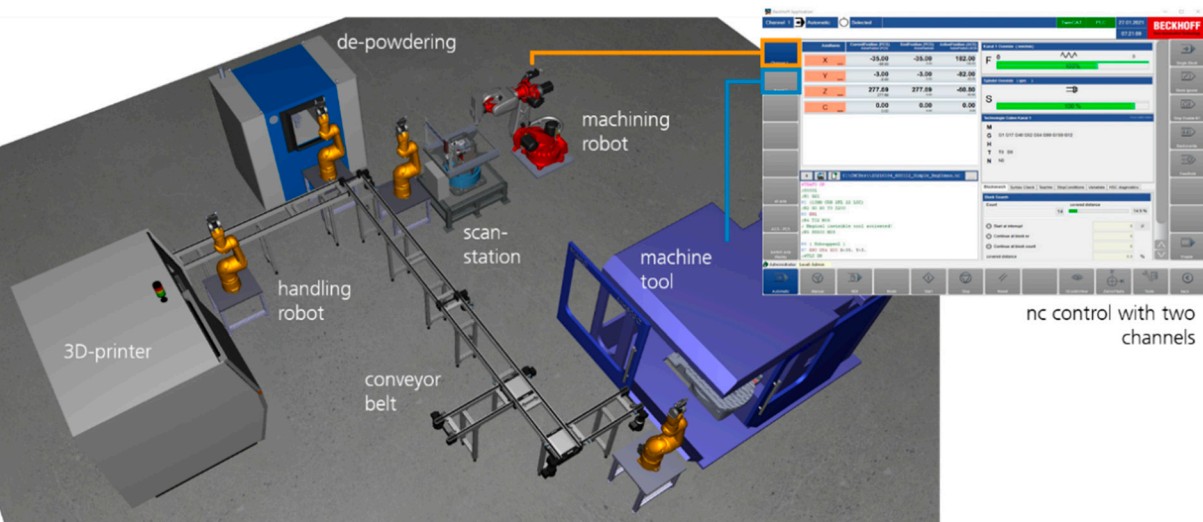

**Figure 14.** Production cell for additive manufacturing with redundant machines for machining finishing.

While the machines perform the same machining task, they have different characteristics, e.g., movement accuracy, stiffness, damping or inertia, which lead to different machining results, as shown in Figure 15. The conventional milling machine has a higher stiffness so that the milling results on the part surface are of higher quality than the results reached with the robot. In the example shown, only the pure vibration behavior of the machines is depicted on the surface of the components. So far, no interactions between the material and the machining tool have been implemented in the model. Thus, the differences could only be evaluated qualitatively until now. Modelling the interactions between process and machine will be part of future work to increase the mapping accuracy in virtual commissioning.

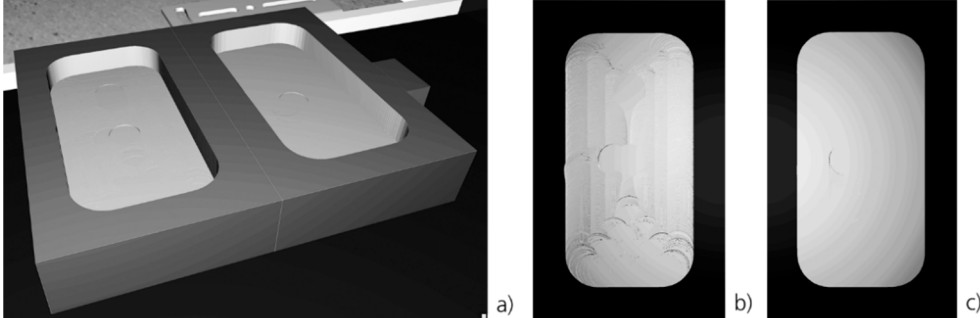

**Figure 15.** (**a**) Comparison of parts manufactured with the same NC Code on different machines. (**b**) Robot, (**c**) machine tool.

The virtual commissioning in combination with physically based models of machines and components allows for the virtual evaluation of reconfiguration strategies before the real production system is redesigned. The individual components of the production system are stored in a library so that a completely free composition of production cells is possible. An example for a matrix production cell is shown in Figure 16. For this model, components from the model depicted in Figure 14 are reused, and a new cell structure is built. In combination with the capability-based system control methods, a fast restructuring of a production system is possible, including its evaluation by use of the digital twin. Depending on the aim of reconfiguration, a variant analysis can be done to evaluate the reachable accuracy of machining or to analyze the system with regard to the production time, the energy consumption or other thinkable optimization criteria. Finally, given the appropriate methods, it can be used to assess the resilience of a production system in the event of unexpected disruptions.

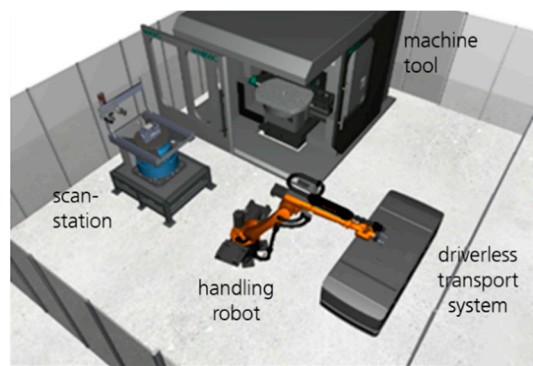

**Figure 16.** Restructured production cell for use in a matrix production system.

## 4. Discussion

### 4.1. Applying Resilient Processes to Production Systems

By amending business process models with contextual information about the involved artifacts, we proposed a solution that allows to the generation and validation of production processes by analyzing the constraints that are imposed by this contextual information. Integrated in an architecture that aims for real-time communication between modular services—including process analysis and process healing—the proposed Process-Planning Engine can be utilized to mitigate risks and to provide resilience to erroneous processes during the execution of the process, which means that error handling strategies do not need to be modelled explicitly beforehand. Instead, "required" and "returned" artifact states can be used to find alternative solutions in case an activity within a process cannot be performed. However, this approach has yet to be verified as part of a real production process since, e.g., the performance of the proposed algorithm or its reliability in generating a suitable "healing process" are crucial factors for the application in a CPPS. Furthermore,

only linear processes are considered at this moment, but production processes often involve parallel processes.

The constraints between the process activities are currently provided by text annotations in a proprietary format, so it is desirable to use an ontology that defines the structure of these annotations, the artifact conditions, and the required agent and builds upon existing ontologies such as the Common Core Ontologies [35] in order to ease the integration of our approach into other systems. Further aspects that need consideration are production equipment and consumables that are required within a production process and therefore may impose further constraints on resilient process planning. Another important aspect is the integration of human agents into these processes. This requires not only addressing how to present information in a user-friendly way, but also how to extend the skill-based approach used to manage non-human agents to "human skills".

### 4.2. Matrix Production as Alternative Manufacturing Concept of the Future

The growing diversity of product variants and the resulting decrease in batch sizes require the flexibility of production. Both internal and external events can lead to changed requirements or to disruptions in the production process that require an adjustment of production sequences and process chains.

Modular production cells combine technologies with the necessary automation for their implementation to realize a defined process step. The matrix production concept allows for the free interlinking of such production cells and thus dissolves the limitations of today's established line production. A wide variety of product variants can be manufactured on one production structure. The sequence of the process steps to be performed is not predetermined by the structure of the manufacturing system. Each product follows its own path through the needed production cells and defines its own production structure. The duration of a single process step no longer determines the cycle of the entire production. Rather, the process steps are conducted independently of each other so that an infrastructure can be created that is oriented to processes and capacity requirements. However, the implementation of manufacturing cells requires control methods that enable a capability-based description of the machining processes. This is the basis for the rapid reconfiguration of the cell and the efficient adaptation to a new manufacturing task.

The matrix production concept enables production to be quickly adjusted to changing requirements. Flexible reactions to customer demands, prioritization of rush orders, or parallel processing of different orders are possible with the same production technology. The robustness of the system increases when central process steps are available in multiple units. This makes production less susceptible to disruptions, as it is possible to switch to another cell if one fails. For this purpose, it is necessary to develop planning methods which, on the one hand, allow a statement about the most efficient processing of the current task, but also allow an evaluation of the planned process sequences with regard to other criteria such as the achievable accuracy during production or also the energy requirement of the production equipment. The methods of virtual commissioning provide the basic functions for the virtual analysis of the production process, considering the real control technology. In the future, it will be necessary to make the required extensions in order to be able to include the dynamic and energetic properties of the manufacturing systems as well as the interactions of machines and processes.

With the methods presented, it is possible to achieve the necessary mutability of production systems, provide the corresponding planning methods, and thus increase the resilience of these systems.

**Author Contributions:** Conceptualization, M.S., A.H. and K.W.; methodology, M.S. and T.W.; formal analysis, M.S.; investigation, M.S. and T.W.; writing—original draft preparation, T.W., M.S., A.H., C.-C.S., K.W. and S.M.; writing—review and editing, S.I., T.W., M.S., A.H., C.-C.S., K.W. and S.M.; visualization, M.S., T.W. and C.-C.S.; funding acquisition, K.W.; supervision, S.I. All authors have read and agreed to the published version of the manuscript.

**Funding:** This research was partially funded by the German Federal Ministry of Education and Research (BMBF) under grant 01IS18067 B (RESPOND) and the Federal Ministry for Economic Affairs and Energy (BMWI) under grant 20940 BG (SynLApp).

**Institutional Review Board Statement:** Not applicable.

**Informed Consent Statement:** Not applicable.

**Data Availability Statement:** Not applicable.

**Conflicts of Interest:** The authors declare no conflict of interest. The funders had no role in the design of the study; in the collection, analyses, or interpretation of data; in the writing of the manuscript; or in the decision to publish the results.

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
