# Peer review of "Increasing Resilience of Production Systems by Integrated Design"

_applsci, doi:10.3390/app11188457_

Round 1

Reviewer 1 Report

Dear authors,

I suggest to take a look at this:

  1. Line 7 – acronym „OPC UA“ acronym is not explained and is used here for the first time
  2. Line 301 – Reference error (Figure 6)

Is it possible for some of the figures to be more clear (transparent ) : fugures 1, 4, 10 and 14?

At the beginig of your paper you have described and categorized risks in industrial practice so that a concept of reconfigurable and robust production systems can be derived.

How do you connect framework for resilient manufacturing systems described in this paper with all categories of risk described? I suggest underlining the connection and applicability of the framework to all risk categories.

Author Response

Dear Reviewer,

Thank you for your helpful comments regarding our paper.

We have incorporated all the changes you requested and hope to have answered the remaining questions. 

With kindest regards,
the authors 

Reviewer 2 Report

This paper present an useful conceptual framework of resilience in manufacturing ecosystem design. Although I am not esperienced in this specific topic, I consider it as a profitable effort for future empirical investigation.

Not by coincidence this conceptual paper has been developed with the aim to address the main Issue of residence when applied in manufacturing firms.

The outiline is clear and well written, in line with the Journal standards. Therefore I suggest to accept the paper in the current forma.

Author Response

Dear Reviewer,

Thank you for your positive feedback regarding our paper.

We have made some minor revisions to explain the methods more adequately. 

With kindest regards,
the authors 
